# Assessing the relationship between male alcohol consumption and intimate partner violence against women in Sub-Saharan Africa: A dual-dataset secondary analysis of recent DHS surveys

Tilahun Belete Mossie[1]*, Haile Mekonnen Fenta[2,3,4], Meseret Tadesse[1], Animut Tadele[5]

1 Department of Psychiatry, College of Medicine and Health Sciences, Bahir Dar University, Bahir Dar, Ethiopia, 2 Department of Statistics, College of Sciences, Bahir Dar University, Bahir Dar, Ethiopia, 3 Center for Environmental and Respiratory Health Research, Population Health, University of Oulu, Oulu, Finland, 4 Biocenter Oulu, University of Oulu, Oulu, Finland, 5 Department of English Language and Literature, College of Education, Bahir Dar University, Bahir Dar, Ethiopia

* tilahunbe100@gmail.com

## Abstract

### Background

Even if the effects of alcohol use on Intimate Partner Violence (IPV) are well documented, most of the research were from developed countries. Sub Saharan Africa is famous for higher rates of IPV, yet its relationship with partners alcohol consumption is rarely studied. This project focused on assessing the relationship between partners alcohol consumption and IPV against married women, using dual dataset for men and women.

### Methods

We used the recent 2021–2022 Demographic and Health Survey (DHS) dataset for five countries in Sub Saharan Africa. We accessed the dataset on February 13, 2024. The data was anonymous, and there was no any personal identifier. The dataset included 37,465 participants. We used women questionnaire to determine IPV and related factors, and partner's questionnaire for alcohol and related factors. STATA 18 was used for analysis. A generalized linear mixed model (GLMM) used to estimate the effect of partner alcohol consumption and other factors on IPV. We determined the level of significance at p-value of less than.05.

### Result

The prevalence of IPV was 33.6% [32.9–34.2%]; and that of partner alcohol consumption accounted 25% [24.5–25.4%]. Partner alcohol consumption increased the odds of physical, emotional, sexual, and any form of IPV.

**Data availability statement:** The dataset used in this study is attached as a supplementary file.

**Funding:** The author(s) received no specific funding for this work.

**Competing interests:** The authors have declared that no competing interests exist.

**Abbreviation:** DHS, Demographic and Health Survey; EAs, Enumeration Areas; ICC, Intra-Class Correlation; IPV, Intimate Partner Violence; SSA, Sub Saharan Africa.

## Conclusion

Partner alcohol consumption increased the odds of IPV, which suggests that partner inclusive intervention packages might be effective in addressing IPV against women in SSA.

## Background

Abuse or harm to a person who is in an intimate relationship, married or cohabiting, that may take the form of physical, emotional, sexual, or other types of violence is termed as Intimate Partner Violence (IPV). IPV can take different forms. Pushing, slapping, twisting, pulling, punching, kicking, beating, burning, and threatening or attacking with a sharp material or a firearm is considered as physical form of IPV. Emotional type of IPV includes verbal threats, humiliation in front of others, or insults. Forced, threatened or similar acts for sexual intercourse without partner's willing is sexual type of IPV [1,2].

Violence has gained global attention. Ending all forms of violence against women is an agenda in the United Nations Sustainable Development Goals. The United Nations Women office reported that about 245 million girls and women experience IPV every year. Each occurrence of IPV may have a significant and enduring effect on the survivors' lives, their families, children, communities as well as society. The worldwide cost of violence against women is tremendous – assessed to be a minimum of USD 1.5 trillion or roughly 2 percent of global Gross Domestic Product (GDP) [2].

Intimate Partner Violence (IPV) against women has been more common in Sub Saharan Africa (SSA). Prior reports showed that despite integrated interventions against IPV, there has been a lack of marked decrements in its prevalence in the region [3–6]. In our previous article, we indicated a disparity in terms of the magnitude and the type of IPV against women across the region. The paper also indicated great discrepancies across catchment areas in a certain country in the experience of IPV [7]. Identifying the factors associated with IPV may be important to address the target groups. Among the different types of violence, physical violence, emotional violence, and sexual violence have been studied at different times. There have been numerous factors significantly associated with IPV. Sociology-demographic and spouse related issues are commonly described factors that showed significant association [3–6].

Partner use of alcohol reported by the wives has been one of the main significant factors. A recent article stated that in SSA, based on spousal report of alcohol consumption in husbands, partner alcohol consumption has significantly associated with emotional, physical, and sexual violence [8]. This study did not include reports from husbands themselves. Previous studies in SSA region have similar findings that partner alcohol consumption increased the probability of IPV against women [9–14].

Alcohol is one of the depressant groups of psychoactive substances. It can influence one's emotion, cognition/thought/, and behavior. Its consumption can lead to

disruptive behavior such as poor decision-making capacity, poor judgement, reckless behavior, and memory loss [15]. Alcohol consumption has been associated with both verbal aggression and physical aggression [10,11,14,15]. Such effects of alcohol consumption by husbands can directly contribute to higher probability of IPV against women.

A couple of existing evidences revealed that partner alcohol consumption is associated with IPV against women [7,8,16]. These findings, however, rely on wives' proxy reports rather than husbands' direct self-reports. The present study addresses this gap by using paired DHS data from both partners across five countries, providing a more reliable dual-dataset assessment of the relationship.

Despite different interventions to lessen IPV in SSA, its prevalence remains higher. Hence, identifying major contributors to IPV is vital to address the challenge. This can help end IPV against women in the region. Programmers, and policy makers can modify their packages based on the outputs of this study. The main aim of this study was to assess the relationship between husbands' alcohol consumption behavior and spouses' experience of IPV in Sub Saharan Africa.

## Materials and methods

### Study design

**Data:** The data for the analysis came from the Demographic and Health Survey (DHS), nationally representative surveys conducted mainly in low-income countries (https://dhsprogram.com). In DHS, a multistage sampling method was used to select samples for each survey conducted in various countries. Hence, the first step of the sampling procedure involved selecting clusters (enumeration areas (EAs)), followed by systematic sampling of the households within the selected EAs. The number of clusters in the first stage was randomly selected from the list of EAs created in recent population census of each country and the households. From the selected households, women aged 15–49 and men 15–59 years were selected for the study [17]. Most of the recent DHS waves used the Global Positioning Systems (GPS) coordinates (latitude and longitude) of household clusters, which helped collect the Geo-spatial information to identify the central point of each EA. Five SSA countries were selected based on the availability of the variables of interest (alcohol use of husband: based on husband's response). Both the women and men record files of the DHS were used to incorporate some relevant covariates. Moreover, the geographical covariates were extracted from the DHS site and were linked to the original individual DHS data sets through the cluster identifying number. A total of 49 (18 in East, 17 in West, 9 in Central and 5 in southern) countries were in Sub-Saharan Africa. Only five countries, namely Kenya, Tanzania, Burkina Faso, Ivory coast, and Ghana were included in the study. This is because, the other countries didn't include the partner's alcohol consumption variable in the dataset.

### Partner alcohol consumption

In the DHS datasets, the partner alcohol consumption was measured from two questionnaires, women and men questionnaires. The women questionnaire focused on the frequency of perceived drunkenness. And it included the following question on the husband's alcohol consumption: "How often (does/did) he get drunk: often, only sometimes, or never?". This variable assessed the perceived frequency of drunkenness, responses ranged from never (0), sometimes (1), and often (2). The second is from the men questionnaire, it assessed the frequency of alcohol consumption, and included questions about the frequency of alcohol consumption: "How often do you drink alcohol: almost every day, about once a week, or less than once a week?".

The previous researchers in SSA investigated the effects of partner alcohol consumption on Intimate Partner Violence; however, their finding focused only on the women perceived responses rather than the actual experience of the partners. We compiled the data from husband's self-report of alcohol consumption from men's record in the dataset. Women dataset was used to assess the experience of IPV and related factors.

**Variables:** The outcome variable for this study includes any form of physical, emotional, sexual violence which was assessed using women's self-reported responses to the questions depending on the modified Conflict of Tactic Scales of Status [18]. Intimate partner violence (IPV) is defined as any instance where women had experienced at least one event of physical, emotional, or sexual violence since the age of 15 years [18–20]. Table 1 presents the detail definitions for different forms of IPV.

**Key explanatory variable** (s): in this study the partner's alcohol consumption was the key covariate. This variable was measured from the men's questionnaire about the frequency of alcohol consumption: "How often do you drink alcohol: almost every day, about once a week, or less than once a week and never?" and managed into two dichotomy groups yes/no for analysis.

**Independent variables:** The independent variables were extracted based on a review of the literature [7,16,21–23].

## Statistical analysis

We used STATA 18 [24] for data management and statistical analysis. *The data were weighted to ensure representativeness and to provide more accurate statistical estimates. We accounted for the complex survey design features of the DHS—namely sampling weights, clustering, and stratification—in all our analyses to ensure that the effect estimates were unbiased and representative at both national and sub-regional levels. In the pooled analyses, we applied de-normalized sampling weights based on each country's population size and additionally adjusted for country and survey year fixed effects to control for unobserved country-specific factors and temporal trends.*

We used the modified Poisson regression models with robust error variance and reported the results as an adjusted prevalence ratio with a 95% confidence interval. This model modified the Poisson regression model since the odds ratio estimated using the traditional logistic regression from a cross-sectional study may overestimate the relative risk when the outcome is common [1,2] and in case of convergence issues this model performs better in estimating the prevalence ratio [3,4].

A generalized linear mixed model (GLMM) [25–29] was employed to examine the effect of the women, husband, and household characteristics on Intimate Partner Violence (IPV) among reproductive aged (15–49 years) married women across five SSA countries. The GLMM model is:

$$g\left(\mu_{ijk}\right) = logit\left(\mu_{ijk}\right) = \log\left(\frac{\mu_{ijk}}{1 - \mu_{ijk}}\right) = \log\left(\frac{P(y_{ijk} = 1)}{P(y_{ijk} = 0)}\right) = \eta_{ijk}$$

**Table 1. The tool used to measure IPV in the demographic and health surveys.**

| Question/item | IPV type |
|---|---|
| Push you, shake you, or throw something at you? | physical IPV |
| Slap you? | |
| Twist your arm or pull your hair | |
| Punch you with his/her first or with something that could hurt you? | |
| Kick you, drag you, or beat you up? | |
| Try to choke you or burn you on purpose? | |
| Threaten or attack you with a knife, gun or any other weapon? | |
| Physically force you to have sexual intercourse with him even when you did not want to? | Sexual IPV |
| Physically force you to perform any other sexual acts you did not want to? | |
| Force you with threats or in any other way to perform sexual acts you did not want to? | |
| Say or do something to humiliate you in front of others? | Emotional IPV |
| Threaten to hurt or harm you or someone close to you? | |
| Insult you or make you feel bad about yourself? | |

the $\mu_{ijk}$ and 1-$\mu_{ijk}$ are respectively the probability of women experiencing any form of IPV and not experiencing IVP (i = 1, ..., 37,565 women, *j* = 1,..., 122 provinces (districts), *k* = 1, ..., 5 countries) where $\beta_0$ is the log odds of intercept; $\beta_1 ... \beta_m$ are effect sizes of women and household-level covariates [30–32].

The fully unconditional three-level model is written for the i<sup>th</sup> women in the j<sup>th</sup> provinces in the k<sup>th</sup> country at the level I as

$$\text{Level I}: \ \eta_{ijk} = \pi_{ijk} + e_{ijk},$$

$$i = 1, 2, 3, ..., 37,565, \ \ j = 1, 2, 3, ..., 476 \text{ and } k = 1, 2, 3, ..., 26$$

for the j<sup>th</sup> province in the k<sup>th</sup> country at level 2 as

$$\text{Level II}: \ \pi_{ijk} = \beta_{00k} + \gamma_{ijk,}$$

and for the k<sup>th</sup> country at level 3 as

$$\text{Level III}: \ \beta_{00k} = \gamma_{000} + \mu_{ook,}$$

where $e_{ijk}$ is a random effect women effect (level I residual), $\gamma_{0jk}$ is a random effect (level II residual), and $\mu_{00k}$ is a random effect (level III residual). The variance components among women within provinces, among provinces within countries, and countries are symbolized by $\sigma^2$, $\tau_\pi$, and $\tau_\beta$ respectively. Moreover, the intra-class correlation (ICC) for the three-level binary data is given for each of the levels separately

$$ICC_{provinces}(level\ 2) = \frac{\sigma^2_{provinces}\left(\tau_\pi\right)}{\sigma^2_{countries}\left(\tau_\beta\right) + \sigma^2_{provinces}\left(\tau_\pi\right) + \frac{\pi^2}{3}} \ ... ICC\ attributable\ to\ level\ 2$$

$$ICC_{countries}(level\ 3) = \frac{\sigma^2_{countries}\left(\tau_\beta\right)}{\sigma^2_{countries}\left(\tau_\beta\right) + \sigma^2_{provinces}\left(\tau_\pi\right) + \frac{\pi^2}{3}} \ ... ICC\ attributable\ to\ level\ 3$$

We assessed multicollinearity using the Variance Inflation Factor (VIF), with values ranging from 1.12 to 3.65, indicating no significant multicollinearity.

### Statement of ethics

This study used a DHS dataset, and did not collect any primary data directly from the participants. All ethical principles for secondary data apply to this project. Participation in the project was based on each countries DHS data collection protocol. Hence, Institutional Review Board was not required for the study.

### Results

The prevalence of Intimate Partner Violence and husband's alcohol consumption among women in five SSA countries. The prevalence of IPV in the selected areas was 33.6%. The pooled prevalence of sexual, physical, emotional violence among women were 6.6%, 20.2%, and 26.6% respectively. Kenya and Ghana recorded the highest prevalence of sexual and emotional violence, respectively. See details in Table 2.

The overall prevalence of partner's alcohol consumption was 25% [24.5–25.4%] with the least prevalence found in Tanzania 16.1% whilst the highest prevalence was recorded in Burkina Faso, 32.6%.

The proportion of IPV among women whose partner consumed alcohol was 38.2%, whereas it was 31.9% among those whose spouse did not consume alcohol. Physical IPV was higher in participants whose partner consumed alcohol than

**Table 2. Prevalence of Intimate Partner Violence and partner alcohol consumption, Sub-Saharan Africa (n = 16320 couples).**

| Countries | Sexual violence | Physical violence | Emotional violence | IPV | Partner's alcohol consumption |
|---|---|---|---|---|---|
| Burkina Faso | 3.7 [3.2–4.2] | 14.2 [13.2–15.2] | 25.9 [24.7–27.2] | 29.1 [27.8–30.4] | 32.6 [31.5–33.8] |
| Ghana | 8.2 [7.4–9.1] | 18.1 [16.9–19.3] | 32.1 [30.7–33.6] | 37.5 [36.0–39.0] | 28.5 [27.3–29.7] |
| Ivory Coast | 5.1 [4.5–5.8] | 17.0 [15.9–18.1] | 23.9 [22.7–25.2] | 29.3 [27.9–30.7] | 27.2 [26.3–28.2] |
| Kenya | 9.0 [8.1–10.0] | 30.1 [28.6–31.6] | 29.5 [28.0–31.0] | 41.3 [39.6–42.9] | 21.2 [20.5–22.0] |
| Tanzania | 8.4 [7.5–9.4] | 25.1 [23.7–26.6] | 21.7 [20.3–23.1] | 32.9 [31.3–34.5] | 16.1 [15.2–17.1] |
| **Pooled (Overall)** | 6.6 [6.3–7.0] | 20.2 [19.6–20.8] | 26.6 [26.0–27.3] | 33.6 [32.9–34.2] | 25.0 [24.5–25.4] |

NB: The prevalence is presented in percentage with 95% confidence interval.

their counterparts (23.6% Vs 18.9%). Similarly, 31.2% of participants with partner alcohol consumption behavior experienced emotional IPV, which is higher as compared to women whose spouse did not consume alcohol (24.9%). Sexual violence was even more common in participants whose husbands consume alcohol (7.7%) than participants whose partner did not consume alcohol (6.2%).

Across age groups, it was higher among older women (37.86% in 45–49 years old) than younger women (29.2% in 15–24 years old). In addition, higher proportions of IPV were recorded in women with primary level of education, who were engaged in a job, who did not justify beating women, and who fear their husbands with respective values of 39.5%, 35.8%, and 71.4%. From partner related factors higher prevalence of IPV was found in partners with primary level of education (38%), and those with high controlling behavior over their spouses (55.6%). In addition, IPV was more common in women with poor wealth index (36.4%), and in those who had four or children (37%). Details are presented in Table 3.

## The relationship between partner alcohol consumption and other factors

The multivariable regression revealed that partner alcohol consumption was significantly associated with IPV. Women whose spouse consumed alcohol had higher odds of suffering from different forms of IPV than participants whose partners did not consume alcohol: physical (1.35 (1.22, 1.48)), sexual (1.22 (1.05, 1.41)), emotional (1.24 (1.13, 1.35)), and any form of IPV (1.28 (1.18, 1.40)).

The other factors that showed significant association with higher odds of IPV included primary level of education than higher education (1.142 (1.16, 1.75)), having job (1.17 (1.07, 1.17)), high autonomy in making decision (1.37 (1.17, 1.60)), exposure to media 1.13 (1.03, 1.24), afraid of husband (3.60 (3.33, 3.84)), partner's primary level of education (1.32 (1.11, 1.56)), partner's high controlling behavior (2.83 (2.50,3.19)), rural residence (1.14 (1.03, 1.25)), and having four or more children (1.56 (1.32, 1.84)). On the other hand, younger women aged 15–24 years had lower odds of IPV than 45–49 years old women (0.75 (0.63, 0.88)). Details are available in Table 4.

## Discussion

The study examined the relationship between IPV and partner alcohol consumption in selected five countries, SSA. These five countries were selected based on the availability of data on partner alcohol consumption, and the inclusion of recent data sets from the year 2020 onwards. The findings of this study revealed that one-third of women in the selected region experienced at least one form of IPV. Among the different forms of IPV, emotional violence was the most prevalent (26.6%), followed by physical violence (20.2%). Sexual violence was the least reported form of violence in the region, but it constituted a significant proportion (6.6%). Kenya recorded the highest prevalence of IPV (41.3% (39.6–42.9%)) whereas Ivory Coast had relatively the least prevalence (29.3% (27.9–30.7)). Though 29.3% was the least proportion of IPV in SSA, it is one of the top 3 proportions worldwide. Only Oceania (49%) and South Asia (35%) reported highest

**Table 3. IPV, partner alcohol consumption and other explanatory variables, SSA (n = 16320 couples).**

| Variables | | Physical | Sexual | Emotional | IPV |
|---|---|---|---|---|---|
| | Outcome variables | 20.20 | 6.62 | 26.63 | 33.57 |
| Partner's Alcohol consumption | Yes | 23.58 | 7.70 | 31.20 | 38.25*** |
| | No | 18.95 | 6.22 | 24.94 | 31.87 |
| **Women level covariates** | | | | | |
| Age in years | 15-24 | 16.71*** | 6.91 | 22.00 | 29.18** |
| | 25-34 | 20.32*** | 5.91 | 27.11 | 34.02 |
| | 35-44 | 21.48** | 7.00 | 28.59 | 35.35 |
| | 45-49 | 24.96 | 7.50 | 30.85 | 37.85 |
| Education | No Education | 18.14** | 4.91** | 26.45 | 31.86 |
| | Primary | 27.53*** | 8.98 | 29.45 | 39.54** |
| | Secondary | 17.32** | 7.07 | 25.20 | 31.56* |
| | Higher | 12.00 | 5.73 | 19.89 | 25.41 |
| Religion | Christian | 17.66 | 5.44 | 25.56 | 31.33 |
| | Muslim | 25.06 | 7.07 | 29.85 | 38.03 |
| | Traditional and Others | 20.21 | 7.31 | 30.31 | 36.37 |
| Working status | Yes | 21.50** | 7.19 | 28.94*** | 35.82*** |
| | No | 17.71 | 5.55 | 22.24 | 29.29 |
| Women attitude to wife-beating | Does not justify | 28.65 | 7.76 | 32.72 | 42.05 |
| | Justifies wife-beating | 19.71** | 6.56 | 26.28 | 33.09 |
| Women decision making (autonomy) | low autonomy | 19.67 | 6.53 | 27.02 | 33.44 |
| | medium autonomy | 20.17 | 6.34 | 27.44 | 34.15 |
| | high autonomy | 20.72*** | 6.93 | 25.65*** | 33.26*** |
| Exposure to media | Yes | 20.66* | 7.06** | 27.27** | 34.36** |
| | No | 18.91* | 5.42** | 24.88** | 31.39** |
| Women afraid of husband | Never afraid | 11.86*** | 3.61*** | 16.86*** | 22.82*** |
| | Sometimes | 26.34 | 8.06 | 26.63 | 43.10 |
| | Most of the time | 54.62*** | 22.29*** | 61.88*** | 71.37*** |
| **Partner level covariates** | | | | | |
| Partner education | No Education | 17.17 | 5.33 | 25.47 | 30.78 |
| | Primary | 25.81** | 7.93 | 28.48* | 38.05** |
| | Secondary | 19.65** | 7.07 | 27.44 | 33.89** |
| | Higher | 14.07** | 4.96 | 20.83 | 26.79** |
| Partner's controlling behavior | No controlling behavior | 14.54*** | 4.20*** | 19.93*** | 25.80*** |
| | Has controlling behavior | 36.25*** | 13.50*** | 45.66*** | 55.63*** |
| **Household level covariates** | | | | | |
| Place of residence | Urban | 17.48 | 6.19 | 24.25 | 30.07** |
| | Rural | 21.76 | 6.89 | 28.01 | 35.58** |
| Wealth Index | Poorest | 22.22 | 6.62 | 28.04 | 35.40 |
| | Poorer | 21.81 | 7.73** | 28.98 | 36.44 |
| | Middle | 20.70 | 6.57* | 27.27 | 34.37 |
| | Richer | 20.24** | 6.52* | 25.29 | 32.60 |
| | Richest | 14.66 | 5.41* | 22.50 | 27.54 |
| Total number of children | 0 | 12.65*** | 7.21 | 18.93** | 25.45*** |
| | 1-4 | 19.07*** | 6.18 | 25.46** | 32.14*** |
| | 4 or more | 23.18*** | 6.96 | 29.71** | 37.01*** |

*p-value < .05, **p-value < .01, ***p-vale < .001.

**Table 4. The association between IPV and partners alcohol consumption and other factors, SSA (n=16320 couples).**

| Variables | | Physical | Sexual | Emotional | IPV |
|---|---|---|---|---|---|
| Partner Alcohol consumption | Yes | 1.35 (1.22, 1.48)*** | 1.22 (1.05, 1.41)** | 1.24 (1.13, 1.35)*** | 1.28 (1.18, 1.40)*** |
| | No | 1 | 1 | 1 | 1 |
| **Women level covariates** | | | | | |
| Age in years | 15-24 | 0.62 (0.51, 0.76)*** | 0.91 (0.67, 1.23) | 0.68 (0.57, 0.81) | 0.75 (0.63, 0.88)** |
| | 25-34 | 0.75 (0.64, 0.88)*** | 0.82 (0.64, 1.06) | 0.87 (0.75, 1.01) | 0.91 (0.79, 1.05) |
| | 35-44 | 0.76 (0.65, 0.89)** | 0.90 (0.71, 1.15) | 0.91 (0.79, 1.04) | 0.91 (0.79, 1.04) |
| | 45-49 | 1 | 1 | 1 | 1 |
| Education | No Education | 1.47 (1.11, 1.93)** | 0.57 (0.39, 0.84)** | 1.13 (.97, 1.29) | 1.22 (0.98, 1.51) |
| | Primary | 1.82 (1.40, 2.37)*** | 0.83 (0.58, 1.20) | 1.30 (.98, 1.62) | 1.142 (1.16, 1.75)** |
| | Secondary | 1.52 (1.18, 1.96)** | 0.83 (0.59, 1.17) | 1.14 (0.93, 1.35) | 1.24 (1.02, 1.50)* |
| | Higher | 1 | 1 | 1 | 1 |
| Working status | Yes | 1.18 (1.07, 1.31)** | 1.08 (0.93, 1.26) | 1.22 (1.12, 1.34)*** | 1.17 (1.07, 1.17)*** |
| | No | 1 | 1 | 1 | 1 |
| Women attitude to wife-beating | Does not justify it | 1 | 1 | 1 | 1 |
| | Justifies wife-beating | 0.74 (0.61, 0.89)** | 1.11 (0.81, 1.53) | 0.97 (0.81, 1.16) | 0.89 (0.72, 1.02) |
| Women decision making (autonomy) | Low autonomy | 1 | 1 | 1 | 1 |
| | Medium autonomy | 0.94 (0.85, 1.04) | 0.95 (0.81, 1.11) | 0.98 (0.89, 1.06) | 0.99 (0.91, 1.08) |
| | High autonomy | 1.42 (1.19, 1.69)*** | 1.12 (0.86, 1.45) | 1.37 (1.17, 1.60)*** | 1.37 (1.17, 1.60)*** |
| Exposure to media | Yes | 1.18 (1.05, 1.33)* | 1.36 (1.13, 1.64)** | 1.13 (1.02, 1.25)** | 1.13 (1.03, 1.24)** |
| | No | 1 | 1 | 1 | 1 |
| Women afraid of husband | Yes | 4.17 (3.83 4.54)*** | 3.42 (3.01 3.91)*** | 3.46 (3.21, 3.72)*** | 3.60 (3.33, 3.84)*** |
| | No | 1 | 1 | 1 | |
| **Partner level covariates** | | | | | |
| Partner's Education | No Education | 1.23 (0.99, 1.53) | 1.36 (0.98, 1.90) | 1.12 (0.93, 1.35) | 1.14 (0.96, 1.36) |
| | Primary | 1.35 (1.09, 1.66)** | 1.28 (0.94, 1.76) | 1.23 (1.03, 1.48)* | 1.32 (1.11, 1.56)** |
| | Secondary | 1.23 (1.01, 1.49)* | 1.20 (0.90, 1.60) | 1.20 (1.01, 1.41)* | 1.19 (1.02, 1.39)* |
| | Higher | 1 | 1 | 1 | 1 |
| Partner's controlling behavior | No | 1 | 1 | 1 | 1 |
| | Yes | 2.80 (2.55, 3.08)*** | 2.58 (2.25, 3.00)*** | 3.04 (2.80, 3.31)*** | 2.83 (2.50,3.19)*** |
| **Household level covariates** | | | | | |
| Place of residence | Urban | 1 | 1 | 1 | 1 |
| | Rural | 1.01 (0.90. 1.14) | 1.01 (0.84, 1.20) | 1.09 (0.99, 1.21) | 1.14 (1.03, 1.25)** |
| Wealth Index | Poorest | 1.13 (0.91, 1.41) | 1.37 (0.98, 1.92) | 0.99 (0.82, 1.19) | 0.96 (0.80, 1.15) |
| | Poorer | 1.09 (0.88, 1.32) | 1.50 (1.10, 2.04)** | 1.04 (0.88, 1.23) | 1.01 (0.85, 1.18) |
| | Middle | 1.13 (0.94, 1.35 | 1.39 (1.04, 1.84)* | 1.03 (0.88, 1.20) | 1.02 (0.88, 1.18) |
| | Richer | 1.29 (1.09, 1.52)** | 1.34 (1.03, 1.74)* | 0.99 (0.86, 1.14) | 1.05 (0.92, 1.20) |
| | Richest | 1 | 1 | 1 | 1 |
| Number of children | 0 | 1 | 1 | 1 | 1 |
| | 1-4 | 1.55 (1.29, 1.87)*** | 0.85 (0.67, 1.07) | 1.40 (1.20, 1.62)** | 1.32 (1.15, 1.53)*** |
| | 4 or more | 1.89 (1.53, 2.33)*** | 0.99 (0.74, 1.32) | 1.56 (1.31, 1.86)** | 1.56 (1.32, 1.84)*** |
| **Random effects parameters** | | | | | |
| Country | Identity | 0.03 (0.01, 0.05) | 0.36 (0.16, 0.81) | 0.15 (0.05, 0.47) | 0.34 (0.16, 0.73)** |
| | ICC | 0.07 (0.02, 0.25) | 0.04 (0.01, 0.16) | 0.01 (0.01, 0.06) | 0.03 (0.01, 0.13) |
| Districts | Identity | 0.53 (0.26, 1.08) | 0.41 (0.31, 0.55) | 0.46 (0.38, 0.56) | 0.45 (0.39, 0.54)** |
| | ICC | 0.12 (0.05, 0.25) | 0.08 (0.04, 0.16) | 0.07 (0.04, 0.10) | 0.09 (0.05, 0.15) |

*p-value<.05, **p-value<.01, ***p-vale<.001.

percentages across the globe [33]. As compared to our previous finding consisting of 26 countries in SSA (42.6%), there is a decrements in the proportion of IPV [7]. However, the proportion is still a globally high figure. Overall, the programs and interventions in SSA in decreasing and eliminating violence against women did not show a marked progress in terms of the proportion of IPV [5,34,35].

In other words, one in four partners consumed alcohol in the study areas. The highest percentage was recorded at Burkina Faso (32.6% (31.5–33.8%)) yet Tanzania recorded relatively the least prevalence (16.1% (15.2–17.1%)). The current finding is lower than the women's perceived report of partner alcohol consumption (36.3%) in 21 SSA countries. The difference was observed even for a similar country, e.g., in Kenya it was 21.2% in the current finding yet it was 33.4% in the previous report [8]. The difference might be explained from the point that the previous study did not take the real experience from the partners themselves, rather it considered women's perception about their partners. Alcohol consumption is associated with impaired psychomotor function, poor cognitive function, and heightened levels of both verbal and physical aggression [11].

Partner alcohol consumption was associated with higher odds of physical, emotional, sexual, or any form of IPV against women in the current study. Physical violence is more common in women whose partner consumed alcohol than women whose partner did not consume alcohol. This could be explained from the point that alcohol impairs self-control and increases aggressive behaviors, which in turn leads to more violence in intimate relationships [36,37]. Similar prior studies from Ethiopia, Sub Saharan Africa, Nepal, and Papua New Guinea indicated that women whose partners consumed alcohol had higher probability of experiencing physical violence [8,38–41]. A multi-country study in 14 cross-cultural nations reported that parental alcohol consumption increased the odds of IPV [13]. The implication might be that parental alcohol consumption not only increases the probability of violence but also exacerbate severity of the violence [42].

In the relationship between partner alcohol consumption and IPV, we found higher odds of emotional violence in women whose partners consumed alcohol than those whose partner did not consume alcohol. Previous findings also support this evidence [41]. Alcohol and other substance use can trigger aggression leading to emotionally abusive behavior [8,15]. In addition, alcohol affects cognitive function resulting in poor self-control behavior. This might also contribute to emotional and other forms of violence in intimate relationship [43].

There is also a direct relationship between partner alcohol consumption and the experience of sexual violence in the current study. Alcohol consumption by the partners increased the odds of experiencing sexual violence against women in intimate relationship. A prior study based on women's report of partners drinking behavior reported that women whose partners consumed alcohol experienced higher rate of sexual violence [8]. In Brazil, and Papua New Guinea similar findings were reported [16,41]. Alcohol consumption has a major role in both victimization and perpetration related to sexual violence. It could be related to impaired judgment, high level of aggression, and the dynamics of power and control within intimate relationships [43].

In the current study, the partner controlling behavior on women increased the odds of IPV. Such controlling behavior can be associated with alcohol consumption. Thus, the relationship between alcohol consumption and IPV is complex that may be mediated by different other factors. This includes educational level of the couple, number of children, sociocultural contexts, and others.

Besides partner alcohol consumption, we found other factors that showed significant association with IPV. Among women characteristics, age 45–49 years, low level of education, having job, high level of autonomy in making decision, exposure to media, and afraid of husband increased the odds of IPV. Partner level factors with higher probability of IPV include low level of education, and high controlling behavior. Others were rural residence, and having four or more children. Despite the strong socio-cultural role of religion in shaping gender norms and alcohol related behaviors in the region, religious differences in IPV prevalence were not contained in the multivariable analysis. Future studies may examine its mediating or moderating effect.

Moreover, this study relied on the dual data of women and their partners to assess the relationship between partner alcohol consumption and IPV, and this is a new approach in the region. Until August 2024, we only found five countries in

SSA that had available DHS paired data on partner alcohol consumption and IPV. A limitation of this study is that participants who experienced intimate partner violence (IPV) in a previous relationship may report these incidents when describing their current relationship, potentially confounding the findings. Also, we cannot establish a direct cause and effect relationship in such conditions.

## Conclusion

We found a significant association between partner alcohol consumption and IPV. Partner alcohol consumption also increased the odds of physical, emotional, and sexual violence. Alcohol consumption is one of the modifiable factors that requires enrolling partners in IPV activities.

Implications of this study include, policy makers and programmers need to give emphasis on partner inclusive programs and interventions with a focus on alcohol consumption, education, and other modifiable sociocultural practices. Also, the program planners shall enhance their collaboration with the local community organizations to make their intervention contextualized to the local context. Researchers might invest on adapting and testing innovative interventions and evaluating the effectiveness of the interventions.

## Supporting information

**S1 File. Data.**
(CSV)

## Acknowledgments

We are delighted to thank the Demographic and Health Survey (http://www.dhsprogram.com) for providing the dataset for the selected countries. We are also very thankful to our families for their time and hard work to keep us on track.

## Author contributions

**Conceptualization:** Tilahun Belete Mossie, Haile Mekonnen Fenta.

**Data curation:** Tilahun Belete Mossie, Haile Mekonnen Fenta, Meseret Tadesse.

**Formal analysis:** Tilahun Belete Mossie, Haile Mekonnen Fenta.

**Investigation:** Haile Mekonnen Fenta, Meseret Tadesse, Animut Tadele.

**Methodology:** Tilahun Belete Mossie, Haile Mekonnen Fenta, Meseret Tadesse.

**Project administration:** Haile Mekonnen Fenta, Animut Tadele.

**Resources:** Haile Mekonnen Fenta, Animut Tadele.

**Software:** Haile Mekonnen Fenta.

**Validation:** Tilahun Belete Mossie, Haile Mekonnen Fenta, Meseret Tadesse, Animut Tadele.

**Visualization:** Tilahun Belete Mossie, Haile Mekonnen Fenta, Meseret Tadesse, Animut Tadele.

**Writing – original draft:** Tilahun Belete Mossie, Haile Mekonnen Fenta, Meseret Tadesse, Animut Tadele.

**Writing – review & editing:** Tilahun Belete Mossie, Haile Mekonnen Fenta, Meseret Tadesse, Animut Tadele.

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
