## [Decision Letter · Decision Letter 0]

1 May 2025

PONE-D-25-06968Assessing the relationship between Male Alcohol consumption and Intimate Partner Violence against Women in Sub-Saharan Africa: A Dual Dataset AnalysisPLOS ONE

Dear Dr. Mossie,

Thank you for submitting your manuscript to PLOS ONE. After careful consideration, we feel that it has merit but does not fully meet PLOS ONE’s publication criteria as it currently stands. Therefore, we invite you to submit a revised version of the manuscript that addresses the points raised during the review process.

**Your manuscript has been assess by one reviewer, whose comments are appended. We recommend that you consider the reviewer's comments carefully in preparing a revised manuscript and response.**

**Please note that we have only been able to secure a single reviewer to assess your manuscript. We are issuing a decision on your manuscript at this point to prevent further delays in the evaluation of your manuscript. Please be aware that the editor who handles your revised manuscript might find it necessary to invite additional reviewers to assess this work once the revised manuscript is submitted. However, we will aim to proceed on the basis of this single review if possible.**

We look forward to receiving your revised manuscript.

Kind regards,

Jason Morgan

Staff Editor

PLOS ONE

**Journal Requirements:**

1. When submitting your revision, we need you to address these additional requirements. Please ensure that your manuscript meets PLOS ONE's style requirements, including those for file naming. The PLOS ONE style templates can be found at https://journals.plos.org/plosone/s/file?id=wjVg/PLOSOne_formatting_sample_main_body.pdf and https://journals.plos.org/plosone/s/file?id=ba62/PLOSOne_formatting_sample_title_authors_affiliations.pdf 2. Thank you for uploading your study's underlying data set. Unfortunately, the repository you have noted in your Data Availability statement does not qualify as an acceptable data repository according to PLOS's standards. At this time, please upload the minimal data set necessary to replicate your study's findings to a stable, public repository (such as figshare or Dryad) and provide us with the relevant URLs, DOIs, or accession numbers that may be used to access these data. For a list of recommended repositories and additional information on PLOS standards for data deposition, please see https://journals.plos.org/plosone/s/recommended-repositories. 3. In the online submission form, you indicated that “for specific requests, please contact the corresponding author through email to get the data at tilahunbe100@gmail.com.”All PLOS journals now require all data underlying the findings described in their manuscript to be freely available to other researchers, either a. In a public repository, b. Within the manuscript itself, or c. Uploaded as supplementary information.This policy applies to all data except where public deposition would breach compliance with the protocol approved by your research ethics board. If your data cannot be made publicly available for ethical or legal reasons (e.g., public availability would compromise patient privacy), please explain your reasons on resubmission and your exemption request will be escalated for approval. 4. When completing the data availability statement of the submission form, you indicated that you will make your data available on acceptance. We strongly recommend all authors decide on a data sharing plan before acceptance, as the process can be lengthy and hold up publication timelines. Please note that, though access restrictions are acceptable now, your entire data will need to be made freely accessible if your manuscript is accepted for publication. This policy applies to all data except where public deposition would breach compliance with the protocol approved by your research ethics board. If you are unable to adhere to our open data policy, please kindly revise your statement to explain your reasoning and we will seek the editor's input on an exemption. Please be assured that, once you have provided your new statement, the assessment of your exemption will not hold up the peer review process. 5. Please amend either the abstract on the online submission form (via Edit Submission) or the abstract in the manuscript so that they are identical. 6. Your ethics statement should only appear in the Methods section of your manuscript. If your ethics statement is written in any section besides the Methods, please move it to the Methods section and delete it from any other section. Please ensure that your ethics statement is included in your manuscript, as the ethics statement entered into the online submission form will not be published alongside your manuscript. 7. We note that Figure 1 in your submission contain map images which may be copyrighted. All PLOS content is published under the Creative Commons Attribution License (CC BY 4.0), which means that the manuscript, images, and Supporting Information files will be freely available online, and any third party is permitted to access, download, copy, distribute, and use these materials in any way, even commercially, with proper attribution. For these reasons, we cannot publish previously copyrighted maps or satellite images created using proprietary data, such as Google software (Google Maps, Street View, and Earth). For more information, see our copyright guidelines: http://journals.plos.org/plosone/s/licenses-and-copyright. We require you to either present written permission from the copyright holder to publish these figures specifically under the CC BY 4.0 license, or remove the figures from your submission: a. You may seek permission from the original copyright holder of Figure 1 to publish the content specifically under the CC BY 4.0 license.   We recommend that you contact the original copyright holder with the Content Permission Form (http://journals.plos.org/plosone/s/file?id=7c09/content-permission-form.pdf) and the following text:“I request permission for the open-access journal PLOS ONE to publish XXX under the Creative Commons Attribution License (CCAL) CC BY 4.0 (http://creativecommons.org/licenses/by/4.0/). Please be aware that this license allows unrestricted use and distribution, even commercially, by third parties. Please reply and provide explicit written permission to publish XXX under a CC BY license and complete the attached form.” Please upload the completed Content Permission Form or other proof of granted permissions as an "Other" file with your submission. In the figure caption of the copyrighted figure, please include the following text: “Reprinted from [ref] under a CC BY license, with permission from [name of publisher], original copyright [original copyright year].” b. If you are unable to obtain permission from the original copyright holder to publish these figures under the CC BY 4.0 license or if the copyright holder’s requirements are incompatible with the CC BY 4.0 license, please either i) remove the figure or ii) supply a replacement figure that complies with the CC BY 4.0 license. Please check copyright information on all replacement figures and update the figure caption with source information. If applicable, please specify in the figure caption text when a figure is similar but not identical to the original image and is therefore for illustrative purposes only.The following resources for replacing copyrighted map figures may be helpful: USGS National Map Viewer (public domain): http://viewer.nationalmap.gov/viewer/The Gateway to Astronaut Photography of Earth (public domain): http://eol.jsc.nasa.gov/sseop/clickmap/Maps at the CIA (public domain): https://www.cia.gov/library/publications/the-world-factbook/index.html and https://www.cia.gov/library/publications/cia-maps-publications/index.htmlNASA Earth Observatory (public domain): http://earthobservatory.nasa.gov/Landsat: http://landsat.visibleearth.nasa.gov/USGS EROS (Earth Resources Observatory and Science (EROS) Center) (public domain): http://eros.usgs.gov/#Natural Earth (public domain): http://www.naturalearthdata.com/

Reviewers' comments:

Reviewer's Responses to Questions

**Comments to the Author**

1. Is the manuscript technically sound, and do the data support the conclusions?

Reviewer #1: Yes

2. Has the statistical analysis been performed appropriately and rigorously? 

Reviewer #1: No

3. Have the authors made all data underlying the findings in their manuscript fully available?

The PLOS Data policy requires authors to make all data underlying the findings described in their manuscript fully available without restriction, with rare exception (please refer to the Data Availability Statement in the manuscript PDF file). The data should be provided as part of the manuscript or its supporting information, or deposited to a public repository. For example, in addition to summary statistics, the data points behind means, medians and variance measures should be available. If there are restrictions on publicly sharing data—e.g. participant privacy or use of data from a third party—those must be specified.requires authors to make all data underlying the findings described in their manuscript fully available without restriction, with rare exception (please refer to the Data Availability Statement in the manuscript PDF file). The data should be provided as part of the manuscript or its supporting information, or deposited to a public repository. For example, in addition to summary statistics, the data points behind means, medians and variance measures should be available. If there are restrictions on publicly sharing data—e.g. participant privacy or use of data from a third party—those must be specified.requires authors to make all data underlying the findings described in their manuscript fully available without restriction, with rare exception (please refer to the Data Availability Statement in the manuscript PDF file). The data should be provided as part of the manuscript or its supporting information, or deposited to a public repository. For example, in addition to summary statistics, the data points behind means, medians and variance measures should be available. If there are restrictions on publicly sharing data—e.g. participant privacy or use of data from a third party—those must be specified.requires authors to make all data underlying the findings described in their manuscript fully available without restriction, with rare exception (please refer to the Data Availability Statement in the manuscript PDF file). The data should be provided as part of the manuscript or its supporting information, or deposited to a public repository. For example, in addition to summary statistics, the data points behind means, medians and variance measures should be available. If there are restrictions on publicly sharing data—e.g. participant privacy or use of data from a third party—those must be specified.

Reviewer #1: Yes

4. Is the manuscript presented in an intelligible fashion and written in standard English?

Reviewer #1: Yes

5. Review Comments to the Author

**Reviewer #1:** Assessing the relationship between Male Alcohol consumption and Intimate Partner Violence against Women in Sub-Saharan Africa: A Dual Dataset AnalysisAssessing the relationship between Male Alcohol consumption and Intimate Partner Violence against Women in Sub-Saharan Africa: A Dual Dataset AnalysisAssessing the relationship between Male Alcohol consumption and Intimate Partner Violence against Women in Sub-Saharan Africa: A Dual Dataset AnalysisAssessing the relationship between Male Alcohol consumption and Intimate Partner Violence against Women in Sub-Saharan Africa: A Dual Dataset Analysis

This paper addresses an important global issue concerning women. Overall, the paper is well written. Check and correct syntactical errors,

Do not use contractions (see page 10 - didn’t) in a professional paper.

Methodology:

- Data analysis

The sampling procedure requires special procedure called weighted complex sampling. This is important for the estimation of appropriate standard errors because of nested sampling. Why were the authors did not conduct complex sampling weighted analysis?

The study outcomes sexual/physical/emotional and pooled IPV are all binary(proportions)

For regression analysis, I would expect them to conduct logistic regression, not linear models which are used if the outcome variable is a continuous variable. Please clarify.

Need details on the process on how the variables were entered. Were they entered all at once or used stepwise regression option?

- Variables

Although the authors provided the reference, they should provide details on how pooled IPV and pooled alcohol consumption were computed. Did they add all 5 countries and take the average?

Multiple regression model:

How did the authors choose which variables to include in the model? Is it based on the bivariate associations presented in Table 2? Please include degrees of freedom and p-value for each crosstabulation. This is important for the reader to see these values. Or use asterisks to indicate the level of significance. *p<.05; ** p=< .01; *** p<.001

Use variables that have a significant association with IPV at the bi-variate level.

There are some independent variables in the model that might be highly associated (multicollinearity). For example, women afraid of husband and partners controlling behavior; working status and wealth index; Country and districts , etc.

Women might be afraid because of husband’s controlling behavior. It has a direct effect.

Check for multicollinearity.

Results/

Second paragraph… Begin with “Table 2 presents bivariate associations between study variables and the outcome.”

Discussion-

The question on IPV asks about violence experienced since the age of 15 years. It is possible that they might not have experienced violence with the current husband. So, it is likely that they might be reporting incidents occurred with their prior partners/boyfriends or their experiences with other family members (incest).

We cannot establish a direct link to IPV and the prevalence of alcohol consumption of the current partner.

We cannot establish a direct link unless the partner was under the influence when the IPV event occurred. So, the authors need to be careful in claiming direct causation between alcohol and committing violence.

Please indicate this fact under limitations.

Please do not repeat the numbers that are already presented in the tables.

Need to add study limitations section and how they might influence the study findings.

Please include your study implications as you see fit for Sub Sahara African region.

Tables:

Include sample size (N) for all the tables and subgroups.

Table 1: delete % symbol next to each number. Put a note at the bottom of the table saying Note: The prevalence is presented in percentage along with 95% confidence intervals in the parenthesis.

Table 2 : please include degrees of freedom and p-value for each crosstabulation. Or use asterisks to indicate the significance level and add a note at the bottom of the table.

This is important for the reader to see the bivariate association between the study variables and the outcome types of IPV.

Table 3:

CI values are missing for education under emotional IPV.

Table 3 title needs to be changed to reflect what is presented.

“Results from multivariable regression predicting odds of IPV” do not use the term “correlation” correlation is a statistical term measuring association between two continuous variables.

Note at the bottom:

Note: Adjusted odds ratios and 95% Confidence Intervals are presented in parenthesis.

Add a note at the bottom of the table for the asterisks to indicate the significance level.

*p<.05; ** p=< .01; *** p<.001

6. PLOS authors have the option to publish the peer review history of their article (what does this mean?). If published, this will include your full peer review and any attached files.). If published, this will include your full peer review and any attached files.). If published, this will include your full peer review and any attached files.). If published, this will include your full peer review and any attached files.

...

Reviewer #1: **Yes:** Suhasini Ramisetty MiklerSuhasini Ramisetty MiklerSuhasini Ramisetty MiklerSuhasini Ramisetty Mikler

---

## [Author Response · Author response to Decision Letter 1]

28 Jul 2025

Point-by-point response to the reviewer

Dear Dr. Mikler,

Thank you for your insightful and valuable feedback, which has greatly strengthened our article. We have revised the manuscript accordingly and would appreciate your review of the updated document.

1.Data analysis

The sampling procedure requires special procedure called weighted complex sampling. This is important for the estimation of appropriate standard errors because of nested sampling. Why were the authors did not conduct complex sampling weighted analysis?

Response: Thank you for your important question. This approach has been addressed in the analysis, and the results are presented in the results section. We have also clarified this further in the manuscript. Specifically, we included the following statement:

"The data were weighted to ensure representativeness and to provide more accurate statistical estimates. We accounted for the complex survey design features of the DHS—namely sampling weights, clustering, and stratification—in all our analyses to ensure that the effect estimates were unbiased and representative at both national and sub-regional levels. In the pooled analyses, we applied de-normalized sampling weights based on each country’s population size and additionally adjusted for country and survey year fixed effects to control for unobserved country-specific factors and temporal trends”.

2.The study outcomes sexual/physical/emotional and pooled IPV are all binary(proportions)

For regression analysis, I would expect them to conduct logistic regression, not linear models which are used if the outcome variable is a continuous variable. Please clarify.

Response: This isa great point. However, the traditional logistic regression model from a cross-sectional study may overestimate the relative risk when the outcome is common and in case of convergence issues modified Poisson regression model performs better in estimating the prevalence ratio. the description is included in the methodology section with references.

“We used the modified Poisson regression models with robust error variance and reported the results as an adjusted prevalence ratio with a 95% confidence interval. This model modified the Posson regression model since the odds ratio estimated using the traditional logistic regression from a cross-sectional study may overestimate the relative risk when the outcome is common [1, 2] and in case of convergence issues this model performs better in estimating the prevalence ratio [3, 4]”.

3.Need details on the process on how the variables were entered. Were they entered all at once or used stepwise regression option?

Response: we used the modified poisson regression model and hence we entered all variables simultaneously.

- Variables

4.Although the authors provided the reference, they should provide details on how pooled IPV and pooled alcohol consumption were computed. Did they add all 5 countries and take the average?

Response: thank you for your valuable comment. We have modified this section in the methodology part.

The IPV is defined as women who had experienced at least one event of physical, emotional or sexual violence since the age of 15 years [5-7].

Table 1: The tool used to measure IPV in the Demographic and Health Surveys.

Question/item IPV type

Push you, shake you, or throw something at you? physical IPV

Slap you?

Twist your arm or pull your hair

Punch you with his/her first or with something that could hurt you?

Kick you, drag you, or beat you up?

Try to choke you or burn you on purpose?

Threaten or attack you with a knife, gun or any other weapon?

Physically force you to have sexual intercourse with him even when you did not want to? Sexual IPV

Physically force you to perform any other sexual acts you did not want to?

Force you with threats or in any other way to perform sexual acts you did not want to?

Say or do something to humiliate you in front of others? Emotional IPV

Threaten to hurt or harm you or someone close to you?

Insult you or make you feel bad about yourself?

Multiple regression model:

5.How did the authors choose which variables to include in the model? Is it based on the bivariate associations presented in Table 2? Please include degrees of freedom and p-value for each crosstabulation. This is important for the reader to see these values. Or use asterisks to indicate the level of significance. *p<.05; ** p=< .01; *** p<.001

Response: This is a good point. However, we used (answered in no. 3) modified poission regression model and we didn’t use the crude and adjusted binary logistic regression models.

6.Use variables that have a significant association with IPV at the bi-variate level.

There are some independent variables in the model that might be highly associated (multicollinearity). For example, women afraid of husband and partners controlling behavior; working status and wealth index; Country and districts , etc.

Women might be afraid because of husband’s controlling behavior. It has a direct effect.

Check for multicollinearity.

Response: We checked the Variance Inflation Factor (VIF) to the variables and it ranged from 1.12 to 3.65 which reveals there was no multicolinearity effect (157-58).

Results/

7. Second paragraph… Begin with “Table 2 presents bivariate associations between study variables and the outcome.”

Response: thanks again, it is corrected (162).

8. Discussion-

The question on IPV asks about violence experienced since the age of 15 years. It is possible that they might not have experienced violence with the current husband. So, it is likely that they might be reporting incidents occurred with their prior partners/boyfriends or their experiences with other family members (incest). We cannot establish a direct link to IPV and the prevalence of alcohol consumption of the current partner.

We cannot establish a direct link unless the partner was under the influence when the IPV event occurred. So, the authors need to be careful in claiming direct causation between alcohol and committing violence. Please indicate this fact under limitations.

Response: we appreciate it, the published DHS reports showed 3-10% of adults got divorce and established new relationships in the region. Hence, we included a limitation statement at the end of the discussion stating that, “A limitation of this study is that participants who experienced intimate partner violence (IPV) in a previous relationship may report these incidents when describing their current relationship, potentially confounding the findings” (265-68).

9.Please do not repeat the numbers that are already presented in the tables.

Need to add study limitations section and how they might influence the study findings.

Please include your study implications as you see fit for Sub Sahara African region.

Response: we revised it accordingly. Also, we amend implications to the policymakers, programmers and researchers (274-79).

10.Tables:

Include sample size (N) for all the tables and subgroups.

Table 1: delete % symbol next to each number. Put a note at the bottom of the table saying Note: The prevalence is presented in percentage along with 95% confidence intervals in the parenthesis.

Table 2 : please include degrees of freedom and p-value for each crosstabulation. Or use asterisks to indicate the significance level and add a note at the bottom of the table.

This is important for the reader to see the bivariate association between the study variables and the outcome types of IPV.

Table 3: CI values are missing for education under emotional IPV.

Table 3 title needs to be changed to reflect what is presented.

“Results from multivariable regression predicting odds of IPV” do not use the term “correlation” correlation is a statistical term measuring association between two continuous variables.Note at the bottom:

Note: Adjusted odds ratios and 95% Confidence Intervals are presented in parenthesis.

Add a note at the bottom of the table for the asterisks to indicate the significance level.

*p<.05; ** p=< .01; *** p<.001

Response: thanks for the critical review, we revised all the tables accordingly.

References

1. Barros, A.J. and V.N. Hirakata, Alternatives for logistic regression in cross-sectional studies: an empirical comparison of models that directly estimate the prevalence ratio. BMC medical research methodology, 2003. 3: p. 1-13.

2. Tamhane, A.R., et al., Prevalence odds ratio versus prevalence ratio: choice comes with consequences. Statistics in medicine, 2016. 35(30): p. 5730-5735.

3. Yelland, L.N., A.B. Salter, and P. Ryan, Performance of the modified Poisson regression approach for estimating relative risks from clustered prospective data. American journal of epidemiology, 2011. 174(8): p. 984-992.

4. Zou, G., A modified poisson regression approach to prospective studies with binary data. American journal of epidemiology, 2004. 159(7): p. 702-706.

5. Chernet, A.G. and K.T. Cherie, Prevalence of intimate partner violence against women and associated factors in Ethiopia. BMC women's health, 2020. 20(1): p. 1-7.

6. Straus, M.A. and E.M. Douglas, A short form of the Revised Conflict Tactics Scales, and typologies for severity and mutuality. Violence and victims, 2004. 19(5): p. 507-520.

7. Temmerman, M., Research priorities to address violence against women and girls. The Lancet, 2015. 385(9978): p. e38-e40.

---

## [Decision Letter · Decision Letter 1]

16 Nov 2025

PONE-D-25-06968R1

Assessing the relationship between Male Alcohol consumption and Intimate Partner Violence against Women in Sub-Saharan Africa: A Dual Dataset Analysis

PLOS ONE

Dear Dr. Mossie,

Thank you for submitting your manuscript to PLOS ONE. After careful consideration, we feel that it has merit but does not fully meet PLOS ONE’s publication criteria as it currently stands. Therefore, we invite you to submit a revised version of the manuscript that addresses the points raised during the review process.

We look forward to receiving your revised manuscript.

Kind regards,

Adewale Olufemi Ashimi, MBBS, MPH, PhD, FWACS

Academic Editor

PLOS ONE

**Journal Requirements:**

**Additional Editor Comments:**

Under the background section of the study, it would be appreciated if you could elaborate on what is known about alcohol and intimate partner violence in SSA with appropriate references and where this work stands.

- Lastly, you may wish to highlight the absence of religion in interpreting these findings, which is important in the sociocultural context of SSA especially in the countries considered.

Reviewers' comments:

Reviewer's Responses to Questions

**Comments to the Author**

1. If the authors have adequately addressed your comments raised in a previous round of review and you feel that this manuscript is now acceptable for publication, you may indicate that here to bypass the “Comments to the Author” section, enter your conflict of interest statement in the “Confidential to Editor” section, and submit your "Accept" recommendation.

Reviewer #2: All comments have been addressed

2. Is the manuscript technically sound, and do the data support the conclusions?

Reviewer #2: (No Response)

3. Has the statistical analysis been performed appropriately and rigorously?

Reviewer #2: (No Response)

4. Have the authors made all data underlying the findings in their manuscript fully available?

The PLOS Data policy requires authors to make all data underlying the findings described in their manuscript fully available without restriction, with rare exception (please refer to the Data Availability Statement in the manuscript PDF file). The data should be provided as part of the manuscript or its supporting information, or deposited to a public repository. For example, in addition to summary statistics, the data points behind means, medians and variance measures should be available. If there are restrictions on publicly sharing data—e.g. participant privacy or use of data from a third party—those must be specified. requires authors to make all data underlying the findings described in their manuscript fully available without restriction, with rare exception (please refer to the Data Availability Statement in the manuscript PDF file). The data should be provided as part of the manuscript or its supporting information, or deposited to a public repository. For example, in addition to summary statistics, the data points behind means, medians and variance measures should be available. If there are restrictions on publicly sharing data—e.g. participant privacy or use of data from a third party—those must be specified. requires authors to make all data underlying the findings described in their manuscript fully available without restriction, with rare exception (please refer to the Data Availability Statement in the manuscript PDF file). The data should be provided as part of the manuscript or its supporting information, or deposited to a public repository. For example, in addition to summary statistics, the data points behind means, medians and variance measures should be available. If there are restrictions on publicly sharing data—e.g. participant privacy or use of data from a third party—those must be specified. requires authors to make all data underlying the findings described in their manuscript fully available without restriction, with rare exception (please refer to the Data Availability Statement in the manuscript PDF file). The data should be provided as part of the manuscript or its supporting information, or deposited to a public repository. For example, in addition to summary statistics, the data points behind means, medians and variance measures should be available. If there are restrictions on publicly sharing data—e.g. participant privacy or use of data from a third party—those must be specified.

Reviewer #2: Yes

5. Is the manuscript presented in an intelligible fashion and written in standard English?

Reviewer #2: Yes

6. Review Comments to the Author

**Reviewer #2:** (No Response)(No Response)(No Response)(No Response)

7. PLOS authors have the option to publish the peer review history of their article (what does this mean?). If published, this will include your full peer review and any attached files.). If published, this will include your full peer review and any attached files.). If published, this will include your full peer review and any attached files.). If published, this will include your full peer review and any attached files.

**Do you want your identity to be public for this peer review?** For information about this choice, including consent withdrawal, please see our  For information about this choice, including consent withdrawal, please see our  For information about this choice, including consent withdrawal, please see our  For information about this choice, including consent withdrawal, please see our Privacy Policy....

Reviewer #2: No

---

## [Author Response · Author response to Decision Letter 2]

19 Dec 2025

Dear, Editor,

we revised both the manuscript and 'revised manuscript with track changes'.

Thanks for the crucial feedback.

---

## [Decision Letter · Decision Letter 2]

12 Jan 2026

PONE-D-25-06968R2

Assessing the relationship between Male Alcohol consumption and Intimate Partner Violence against Women in Sub-Saharan Africa: A Dual Dataset Analysis

PLOS One

Dear Dr.  Mossie,

Thank you for submitting your manuscript to PLOS ONE. After careful consideration, we feel that it has merit but does not fully meet PLOS ONE’s publication criteria as it currently stands. Therefore, we invite you to submit a revised version of the manuscript that addresses the points raised during the review process.

**ACADEMIC EDITOR:**

According to the STROBE reporting guideline, the study design should be in the title. Indicate the study design in the title of the manuscript. You may wish to rewrite the title.

We look forward to receiving your revised manuscript.

Kind regards,

Adewale Olufemi Ashimi, MBBS, MPH, PhD, FWACS

Academic Editor

PLOS One

Journal Requirements:

Reviewer's Responses to Questions

**Comments to the Author**

1. If the authors have adequately addressed your comments raised in a previous round of review and you feel that this manuscript is now acceptable for publication, you may indicate that here to bypass the “Comments to the Author” section, enter your conflict of interest statement in the “Confidential to Editor” section, and submit your "Accept" recommendation.

Reviewer #1: All comments have been addressed

Reviewer #2: All comments have been addressed

2. Is the manuscript technically sound, and do the data support the conclusions?

Reviewer #1: Yes

Reviewer #2: (No Response)

3. Has the statistical analysis been performed appropriately and rigorously? 

Reviewer #1: (No Response)

Reviewer #2: (No Response)

4. Have the authors made all data underlying the findings in their manuscript fully available?

The PLOS Data policy requires authors to make all data underlying the findings described in their manuscript fully available without restriction, with rare exception (please refer to the Data Availability Statement in the manuscript PDF file). The data should be provided as part of the manuscript or its supporting information, or deposited to a public repository. For example, in addition to summary statistics, the data points behind means, medians and variance measures should be available. If there are restrictions on publicly sharing data—e.g. participant privacy or use of data from a third party—those must be specified.requires authors to make all data underlying the findings described in their manuscript fully available without restriction, with rare exception (please refer to the Data Availability Statement in the manuscript PDF file). The data should be provided as part of the manuscript or its supporting information, or deposited to a public repository. For example, in addition to summary statistics, the data points behind means, medians and variance measures should be available. If there are restrictions on publicly sharing data—e.g. participant privacy or use of data from a third party—those must be specified.requires authors to make all data underlying the findings described in their manuscript fully available without restriction, with rare exception (please refer to the Data Availability Statement in the manuscript PDF file). The data should be provided as part of the manuscript or its supporting information, or deposited to a public repository. For example, in addition to summary statistics, the data points behind means, medians and variance measures should be available. If there are restrictions on publicly sharing data—e.g. participant privacy or use of data from a third party—those must be specified.requires authors to make all data underlying the findings described in their manuscript fully available without restriction, with rare exception (please refer to the Data Availability Statement in the manuscript PDF file). The data should be provided as part of the manuscript or its supporting information, or deposited to a public repository. For example, in addition to summary statistics, the data points behind means, medians and variance measures should be available. If there are restrictions on publicly sharing data—e.g. participant privacy or use of data from a third party—those must be specified.

Reviewer #1: (No Response)

Reviewer #2: (No Response)

5. Is the manuscript presented in an intelligible fashion and written in standard English?

Reviewer #1: Yes

Reviewer #2: (No Response)

6. Review Comments to the Author

Reviewer #1: PLOS one paper review December 2025/Jan 2026 REVISION#2

Thank you for considering my review comments on the previous version/s.

Here are some minor comments you might want to consider:

Abstract:

I think the authors should indicate that the final analysis includes 16320 couples instead of 37,465 participants.

REORGANIZE THE TEXT IN METHODOLOGY FOR A BETTER FLOW:

Describe the outcome of interest first

Change “Variables” subheading to “Outcome variable”

DeleteTable 1- include that text in the outcome variable description

Move the text lines 106 to 120 to under Key explanatory variable because alcohol consumption is your main predictor of interest.

Statistical Analysis:

Then the authors need to explain the process how they decided to include the final variables (independent predictors) in the regression model.

How did the authors choose which variables to include in the model? The authors failed to address this

Wealth index and religion are not associated at the bivariate level, so they must be excluded in the final model. Adding too many/unnecessary variables in the final model leads to Overfitting and the model appears to fit well to the current data.

There are some independent variables in the model that might be highly associated (multicollinearity). For example, women afraid of husband and partners controlling behavior;

Briefly report your multicollinearity analysis results to address this concern.

This is still a concern in their analysis. I strongly suggest the authors to rerun the models by excluding these variables and test their assumptions.

Results:

The authors can still reduce the repetition of numbers that are presented in the tables. Just highlight the main findings in this section.

Reviewer #2: (No Response)

7. PLOS authors have the option to publish the peer review history of their article (what does this mean?). If published, this will include your full peer review and any attached files.). If published, this will include your full peer review and any attached files.). If published, this will include your full peer review and any attached files.). If published, this will include your full peer review and any attached files.

...

Reviewer #1: No

Reviewer #2: No

---

## [Author Response · Author response to Decision Letter 3]

31 Mar 2026

Additional Editor Comments:

Title: we revised the title according to the PLOS policy to, “Assessing the relationship between Male Alcohol consumption and Intimate Partner Violence against Women in Sub-Saharan Africa: A Dual-Dataset Secondary Analysis of Recent DHS Surveys”

Please review your reference list to ensure that it is complete and correct.

Response: after careful review, we have revised the references by removing two duplicated references, the prior references 43 and 45 were duplicates of 15, and 21. Hence, we have revised it.

Under the background section of the study, it would be appreciated if you could elaborate on what is known about alcohol and intimate partner violence in SSA with appropriate references and where this work stands.

Response: we have revised the manuscript by including a statement describing this feedback (line 75 – 78);

existing evidence revealed that partner alcohol consumption is associated with IPV against women [7, 8 21]. These findings, however, rely on wives’ proxy reports rather than husbands’ direct self-reports. The present study addresses this gap by using paired DHS data from both partners across five countries, providing a more reliable dual-dataset assessment of the relationship

Lastly, you may wish to highlight the absence of religion in interpreting these findings, which is important in the sociocultural context of SSA especially in the countries considered.

Response: again, thanks for the relevant feedback, we have indicated it in the discussion section (line 293 – 300)

Despite the strong socio-cultural role of religion in shaping gender norms and alcohol related behaviors in the region, religious differences in IPV prevalence were not contained in the multivariable analysis. Future studies may examine its mediating or moderating effect.

Point-by-point response to the reviewer

Dear Dr. Mikler,

Thank you for your insightful and valuable feedback, which has greatly strengthened our article. We have revised the manuscript accordingly and would appreciate your review of the updated document.

1. Data analysis

The sampling procedure requires special procedure called weighted complex sampling. This is important for the estimation of appropriate standard errors because of nested sampling. Why were the authors did not conduct complex sampling weighted analysis?

Response: Thank you for your important question. This approach has been addressed in the analysis, and the results are presented in the results section. We have also clarified this further in the manuscript. Specifically, we included the following statement:

"The data were weighted to ensure representativeness and to provide more accurate statistical estimates. We accounted for the complex survey design features of the DHS—namely sampling weights, clustering, and stratification—in all our analyses to ensure that the effect estimates were unbiased and representative at both national and sub-regional levels. In the pooled analyses, we applied de-normalized sampling weights based on each country’s population size and additionally adjusted for country and survey year fixed effects to control for unobserved country-specific factors and temporal trends”.

2. The study outcomes sexual/physical/emotional and pooled IPV are all binary(proportions)

For regression analysis, I would expect them to conduct logistic regression, not linear models which are used if the outcome variable is a continuous variable. Please clarify.

Response: This isa great point. However, the traditional logistic regression model from a cross-sectional study may overestimate the relative risk when the outcome is common and in case of convergence issues modified Poisson regression model performs better in estimating the prevalence ratio. the description is included in the methodology section with references.

“We used the modified Poisson regression models with robust error variance and reported the results as an adjusted prevalence ratio with a 95% confidence interval. This model modified the Posson regression model since the odds ratio estimated using the traditional logistic regression from a cross-sectional study may overestimate the relative risk when the outcome is common [1, 2] and in case of convergence issues this model performs better in estimating the prevalence ratio [3, 4]”.

3. Need details on the process on how the variables were entered. Were they entered all at once or used stepwise regression option?

Response: we used the modified poisson regression model and hence we entered all variables simultaneously.

- Variables

4. Although the authors provided the reference, they should provide details on how pooled IPV and pooled alcohol consumption were computed. Did they add all 5 countries and take the average?

Response: thank you for your valuable comment. We have modified this section in the methodology part.

The IPV is defined as women who had experienced at least one event of physical, emotional or sexual violence since the age of 15 years [5-7].

Table 1: The tool used to measure IPV in the Demographic and Health Surveys.

Question/item IPV type

Push you, shake you, or throw something at you? physical IPV

Slap you?

Twist your arm or pull your hair

Punch you with his/her first or with something that could hurt you?

Kick you, drag you, or beat you up?

Try to choke you or burn you on purpose?

Threaten or attack you with a knife, gun or any other weapon?

Physically force you to have sexual intercourse with him even when you did not want to? Sexual IPV

Physically force you to perform any other sexual acts you did not want to?

Force you with threats or in any other way to perform sexual acts you did not want to?

Say or do something to humiliate you in front of others? Emotional IPV

Threaten to hurt or harm you or someone close to you?

Insult you or make you feel bad about yourself?

Multiple regression model:

5. How did the authors choose which variables to include in the model? Is it based on the bivariate associations presented in Table 2? Please include degrees of freedom and p-value for each crosstabulation. This is important for the reader to see these values. Or use asterisks to indicate the level of significance. *p<.05; ** p=< .01; *** p<.001

Response: This is a good point. However, we used (answered in no. 3) modified poission regression model and we didn’t use the crude and adjusted binary logistic regression models.

6. Use variables that have a significant association with IPV at the bi-variate level.

There are some independent variables in the model that might be highly associated (multicollinearity). For example, women afraid of husband and partners controlling behavior; working status and wealth index; Country and districts , etc.

Women might be afraid because of husband’s controlling behavior. It has a direct effect.

Check for multicollinearity.

Response: We checked the Variance Inflation Factor (VIF) to the variables and it ranged from 1.12 to 3.65 which reveals there was no multicolinearity effect (157-58).

Results/

7. Second paragraph… Begin with “Table 2 presents bivariate associations between study variables and the outcome.”

Response: thanks again, it is corrected (162).

8. Discussion-

The question on IPV asks about violence experienced since the age of 15 years. It is possible that they might not have experienced violence with the current husband. So, it is likely that they might be reporting incidents occurred with their prior partners/boyfriends or their experiences with other family members (incest). We cannot establish a direct link to IPV and the prevalence of alcohol consumption of the current partner.

We cannot establish a direct link unless the partner was under the influence when the IPV event occurred. So, the authors need to be careful in claiming direct causation between alcohol and committing violence. Please indicate this fact under limitations.

Response: we appreciate it, the published DHS reports showed 3-10% of adults got divorce and established new relationships in the region. Hence, we included a limitation statement at the end of the discussion stating that, “A limitation of this study is that participants who experienced intimate partner violence (IPV) in a previous relationship may report these incidents when describing their current relationship, potentially confounding the findings” (265-68).

9. Please do not repeat the numbers that are already presented in the tables.

Need to add study limitations section and how they might influence the study findings.

Please include your study implications as you see fit for Sub Sahara African region.

Response: we revised it accordingly. Also, we amend implications to the policymakers, programmers and researchers (274-79).

10. Tables:

Include sample size (N) for all the tables and subgroups.

Table 1: delete % symbol next to each number. Put a note at the bottom of the table saying Note: The prevalence is presented in percentage along with 95% confidence intervals in the parenthesis.

Table 2 : please include degrees of freedom and p-value for each crosstabulation. Or use asterisks to indicate the significance level and add a note at the bottom of the table.

This is important for the reader to see the bivariate association between the study variables and the outcome types of IPV.

Table 3: CI values are missing for education under emotional IPV.

Table 3 title needs to be changed to reflect what is presented.

“Results from multivariable regression predicting odds of IPV” do not use the term “correlation” correlation is a statistical term measuring association between two continuous variables.Note at the bottom:

Note: Adjusted odds ratios and 95% Confidence Intervals are presented in parenthesis.

Add a note at the bottom of the table for the asterisks to indicate the significance level.

*p<.05; ** p=< .01; *** p<.001

Response: thanks for the critical review, we revised all the tables accordingly.

References

1. Barros, A.J. and V.N. Hirakata, Alternatives for logistic regression in cross-sectional studies: an empirical comparison of models that directly estimate the prevalence ratio. BMC medical research methodology, 2003. 3: p. 1-13.

2. Tamhane, A.R., et al., Prevalence odds ratio versus prevalence ratio: choice comes with consequences. Statistics in medicine, 2016. 35(30): p. 5730-5735.

3. Yelland, L.N., A.B. Salter, and P. Ryan, Performance of the modified Poisson regression approach for estimating relative risks from clustered prospective data. American journal of epidemiology, 2011. 174(8): p. 984-992.

4. Zou, G., A modified poisson regression approach to prospective studies with binary data. American journal of epidemiology, 2004. 159(7): p. 702-706.

5. Chernet, A.G. and K.T. Cherie, Prevalence of intimate partner violence against women and associated factors in Ethiopia. BMC women's health, 2020. 20(1): p. 1-7.

6. Straus, M.A. and E.M. Douglas, A short form of the Revised Conflict Tactics Scales, and typologies for severity and mutuality. Violence and victims, 2004. 19(5): p. 507-520.

7. Temmerman, M., Research priorities to address violence against women and girls. The Lancet, 2015. 385(9978): p. e38-e40.

---

## [Decision Letter · Decision Letter 3]

3 Apr 2026

Assessing the relationship between Male Alcohol consumption and Intimate Partner Violence against Women in Sub-Saharan Africa: A Dual-Dataset Secondary Analysis of Recent DHS Surveys

PONE-D-25-06968R3

Dear Mr. Mossie,

We’re pleased to inform you that your manuscript has been judged scientifically suitable for publication and will be formally accepted for publication once it meets all outstanding technical requirements.

Kind regards,

Adewale Olufemi Ashimi, MBBS, MPH, PhD, FWACS

Academic Editor

PLOS One

Additional Editor Comments (optional):

Reviewers' comments:

Reviewer's Responses to Questions

**Comments to the Author**

1. If the authors have adequately addressed your comments raised in a previous round of review and you feel that this manuscript is now acceptable for publication, you may indicate that here to bypass the “Comments to the Author” section, enter your conflict of interest statement in the “Confidential to Editor” section, and submit your "Accept" recommendation.

Reviewer #2: All comments have been addressed

2. Is the manuscript technically sound, and do the data support the conclusions?

Reviewer #2: (No Response)

3. Has the statistical analysis been performed appropriately and rigorously? 

Reviewer #2: (No Response)

4. Have the authors made all data underlying the findings in their manuscript fully available?

The PLOS Data policy requires authors to make all data underlying the findings described in their manuscript fully available without restriction, with rare exception (please refer to the Data Availability Statement in the manuscript PDF file). The data should be provided as part of the manuscript or its supporting information, or deposited to a public repository. For example, in addition to summary statistics, the data points behind means, medians and variance measures should be available. If there are restrictions on publicly sharing data—e.g. participant privacy or use of data from a third party—those must be specified.requires authors to make all data underlying the findings described in their manuscript fully available without restriction, with rare exception (please refer to the Data Availability Statement in the manuscript PDF file). The data should be provided as part of the manuscript or its supporting information, or deposited to a public repository. For example, in addition to summary statistics, the data points behind means, medians and variance measures should be available. If there are restrictions on publicly sharing data—e.g. participant privacy or use of data from a third party—those must be specified.requires authors to make all data underlying the findings described in their manuscript fully available without restriction, with rare exception (please refer to the Data Availability Statement in the manuscript PDF file). The data should be provided as part of the manuscript or its supporting information, or deposited to a public repository. For example, in addition to summary statistics, the data points behind means, medians and variance measures should be available. If there are restrictions on publicly sharing data—e.g. participant privacy or use of data from a third party—those must be specified.requires authors to make all data underlying the findings described in their manuscript fully available without restriction, with rare exception (please refer to the Data Availability Statement in the manuscript PDF file). The data should be provided as part of the manuscript or its supporting information, or deposited to a public repository. For example, in addition to summary statistics, the data points behind means, medians and variance measures should be available. If there are restrictions on publicly sharing data—e.g. participant privacy or use of data from a third party—those must be specified.

Reviewer #2: (No Response)

5. Is the manuscript presented in an intelligible fashion and written in standard English?

Reviewer #2: (No Response)

6. Review Comments to the Author

Reviewer #2: (No Response)

7. PLOS authors have the option to publish the peer review history of their article (what does this mean?). If published, this will include your full peer review and any attached files.). If published, this will include your full peer review and any attached files.). If published, this will include your full peer review and any attached files.). If published, this will include your full peer review and any attached files.

...

Reviewer #2: No

---

## [Editor Report · Acceptance letter]

PONE-D-25-06968R3

PLOS One

Dear Dr. Mossie,

I'm pleased to inform you that your manuscript has been deemed suitable for publication in PLOS One. Congratulations! Your manuscript is now being handed over to our production team.

Kind regards,

on behalf of

Dr. Adewale Olufemi Ashimi

Academic Editor

PLOS One